# The Roles of Sodium-Independent Inorganic Phosphate Transporters in Inorganic Phosphate Homeostasis and in Cancer and Other Diseases

**DOI:** 10.3390/ijms21239298

**Published:** 2020-12-06

**Authors:** Marco Antonio Lacerda-Abreu, Thais Russo-Abrahão, Jose Roberto Meyer-Fernandes

**Affiliations:** 1Laboratório de Bioquímica Celular, Instituto de Bioquímica Médica Leopoldo de Meis, Universidade Federal do Rio de Janeiro, Rio de Janeiro, RJ 21941-901, Brazil; marcoantoniolacerdaabreu@gmail.com (M.A.L.-A.); trusso@bioqmed.ufrj.br (T.R.-A.); 2Instituto Nacional de Ciência e Tecnologia em Biologia Estrutural e Bioimagem, Rio de Janeiro, RJ 21941-590, Brazil

**Keywords:** inorganic phosphate, sodium-independent Pi transport, inorganic phosphate homeostasis, cancer

## Abstract

Inorganic phosphate (Pi) is an essential nutrient for the maintenance of cells. In healthy mammals, extracellular Pi is maintained within a narrow concentration range of 0.70 to 1.55 mM. Mammalian cells depend on Na^+^/Pi cotransporters for Pi absorption, which have been well studied. However, a new type of sodium-independent Pi transporter has been identified. This transporter assists in the absorption of Pi by intestinal cells and renal proximal tubule cells and in the reabsorption of Pi by osteoclasts and capillaries of the blood–brain barrier (BBB). Hyperphosphatemia is a risk factor for mineral deposition, the development of diseases such as osteoarthritis, and vascular calcifications (VCs). Na^+^-independent Pi transporters have been identified and biochemically characterized in vascular smooth muscle cells (VSMCs), chondrocytes, and matrix vesicles, and their involvement in mineral deposition in the extracellular microenvironment has been suggested. According to the growth rate hypothesis, cancer cells require more phosphate than healthy cells due to their rapid growth rates. Recently, it was demonstrated that breast cancer cells (MDA-MB-231) respond to high Pi concentration (2 mM) by decreasing Na^+^-dependent Pi transport activity concomitant with an increase in Na^+^-independent (H^+^-dependent) Pi transport. This Pi H^+^-dependent transport has a fundamental role in the proliferation and migratory capacity of MDA-MB-231 cells. The purpose of this review is to discuss experimental findings regarding Na^+^-independent inorganic phosphate transporters and summarize their roles in Pi homeostasis, cancers and other diseases, such as osteoarthritis, and in processes such as VC.

## 1. Introduction

Inorganic phosphate (Pi) is an essential nutrient for the formation of ATP skeletal mineralization and the constituents of DNA, RNA, phospholipids and a variety of phosphorylated metabolic intermediates [1,2]. Due to the ability of Pi to donate H^+^ through OH^-^ groups, it contributes to the buffer system in the blood (pH 7.4), modulating the ratio of monovalent to bivalent phosphate based on pH [3].

Pi absorption is possible due to transporter-mediated translocation across cell membranes. Pi absorption through Na^+^/Pi cotransporters has been established in mammalian cells. These cotransporters constitute two large families of inorganic phosphate transporters: SLC20 and SLC34 [4,5]. The SLC20 family comprises two members, i.e., PiT-1 (encoded by SLC20A1) and PiT-2 (encoded by SLC20A2), both of which are sodium–phosphate cotransporters that preferably carry monovalent inorganic phosphate (H_2_PO_4_^−^) together with two sodium ions. These transporters are expressed almost exclusively in the kidney [4,5].

The SLC34 family contains three members, namely NaPi-IIa (SLC34A1), NaPi-IIb (SLC34A2) and NaPi-IIc (SLC34A3), all of which are sodium–phosphate cotransporters; however, they vary in their biochemical kinetics. These proteins transport divalent inorganic phosphate (HPO_4_^2−^) together with two or three sodium ions [5,6]. A Na^+^-dependent Pi transporter family (NaPi-I, NPTI) also exists, though it also carries anions other than Pi. NPTI is predominantly expressed in the proximal brush border tubular membrane and acts more as an intrinsic Pi transport modulator than as a Na^+^/Pi cotransporter [7,8].

Although Pi Na^+^-dependent transporters have been investigated in detail, another class of sodium-independent Pi transporters has been identified in intestinal [9] and renal cells [10], osteoclasts [11], chondrocytes [12], endothelial cells [13], blood–brain barrier (BBB) capillaries [14] and tumor cells [15]. Studies have reported the kinetic component as being Na^+^-independent with a Pi transporter, not as a nonspecific Pi transport, showing that it is not merely a consequence of increased diffusion. This transport exhibits specific characteristics of saturation, pH dependence and inhibition, and along with sodium-dependent Pi, plays a significant role in Pi homeostasis.

In healthy mammals, extracellular Pi is maintained within a narrow range of concentrations: between 0.70 and 1.55 mM [16,17]. Phosphate metabolism in the body involves complex interactions among several factors that can affect the digestion, absorption, distribution and excretion of this element, maintaining Pi homeostasis [8]. In this work, we review the mechanisms of Na^+^-independent phosphate transporters present in some tissues and discuss their possible roles in the regulation of Pi homeostasis and the development of some diseases.

## 2. Na^+^-Independent P_i_ Transport System for P_i_ Homeostasis

### 2.1. Pi Transport System in Intestinal Pi Absorption

The physiological luminal phosphate concentrations in the human jejunum range from 0.7 to 12.7 mM, depending on the food type ingested [18]. However, it has been demonstrated that phosphate uptake is critically dependent on sodium, which has an affinity for phosphate of ~0.1 mmol/L and is responsible for only 50% of transepithelial phosphate transport across the small intestine. The remaining 50% of transepithelial phosphate transport is Na^+^-independent (Figure 1) [19].

A study using the human cell line Caco2BBE as a model of intestinal cells was performed to biochemically characterize the transport of Pi of cells grown at 1 mM Pi (control condition) or 4 mM Pi, the latter representing the high concentration of Pi in the intestinal lumen [9]. The cells in all conditions exhibited Na^+^-independent Pi transport, suggesting that type II and III Pi transporters are not involved. At 1 mM Pi, Na^+^-independent Pi transport was proton-activated, as evidenced by its inhibition by two proton ionophores (Carbonyl cyanide-4-(trifluoromethoxy)phenylhydrazone-FCCP and Carbonyl cyanide- m-chlorophenyl hydrazine-CCCP) [9]. In addition, increased Pi uptake was observed to be insensitive to FCCP and CCCP at 4 mM Pi (Table 1) [9].

Candeal et al. attempted to identify potential transporters involved in Na^+^-independent Pi transport in several models by analyzing either RNA expression differences between cells maintained at 1 or 4 mM Pi or the siRNA-induced downregulation of specific transporters. None of these approaches were successful, but they helped to eliminate SLC26, SLC20, SLC17 and SLC34 family members as possible candidates [9].

In other animal models, Na^+^-independent Pi transport has been studied; however, no conclusions about sodium-independent Pi transporters have been possible, although much speculation has been offered. In intestinal brush border membrane vesicles (BBMVs) prepared from rat jejunum, the Na^+^-independent, diffusional component of intestinal Pi transport represents approximately 40–50% of the total uptake [21]. This component is significantly higher in the rat ileum than in the rat jejunum. Intestinal Pi transport occurs by both a sodium-independent, unsaturated process and an active, sodium-dependent component of phosphate absorption, mainly in the duodenum and jejunum [21].

The relative contributions of Na^+^-dependent and Na^+^-independent phosphate transport along the rat intestine have been characterized using duodenal, jejunal and ileal regions of the small intestine and proximal and distal colon. These contributions suggest that transepithelial phosphate absorption in vivo is predominately Na^+^-independent, with Na^+^-dependent (presumably NaPi-IIb-mediated) transport playing a lesser role than currently thought [22]. The authors responsible for these findings discussed two possibilities regarding sodium-independent Pi transport: (1) some tight junction proteins, such as claudin, might function as channels that provide a selective paracellular route for passive ion flow. Apparently, it plays a role in the absorption of calcium in the small intestine, but paracellular phosphate transport has not been investigated [22,23]. (2) A yet-to-be identified pathway for Na^+^-independent phosphate transport may exist [22].

Another research group studying the small intestine of rats found substantial sodium-independent Pi uptake. The authors responsible for this finding argue that at least some of this sodium-independent Pi transport could be explained by the activity of PiT-2, as this transporter is still capable of functioning in the absence of sodium, although with minimal activity [24]; however, this hypothesis has not yet been tested.

Some studies of ruminant intestines have shown the additional participation of Pi Na^+^-independent transporters in Pi absorption [25]. Shirazi-Beechey et al. showed the existence of H^+^- rather than Na^+^-coupling for Pi transport in the ruminant intestine [26]. This finding was achieved by the use of compounds that are capable of dissipating the imposed H^+^ gradient (4,5,6,7-tetrachloro-2-trifluoromethylbenzimidazole, TTFB) and inhibiting the uptake of phosphate into ovine duodenal brush-border membrane vesicles. The pH at the site of Pi transport in the ovine intestine is 3 to 4. At this pH range, the majority (98%) of the orthophosphate is in the monovalent form, suggesting that the species to be transported is H_2_PO_4_ [26]. In goats, it was demonstrated that two different transport systems mediate intestinal Pi transport. In goat duodenum, an H^+^-dependent and Na^+^-sensitive Pi transport system was identified that does not belong to the NaPi type II family. In contrast, in the goat jejunum, Na^+^-dependent, H^+^-sensitive Pi transport is mainly mediated by NaPi-IIb [27].

### 2.2. Bone Resorption and Pi Transport Coupled to the Proton Gradient

The skeleton is continually changing in its mass or form via the activities of osteoblasts (the cells responsible for bone formation) and osteoclasts (the cells responsible for bone resorption) [28]. Bone resorption depends on the ability of osteoclasts to generate extracellular acid compartments containing vacuolar-type H^+^-adenosine triphosphatase (V-type ATPase), which degrades hydroxyapatite ([Ca_3_(PO_4_)2]3Ca(OH)_2_) into Ca^+^, water and phosphate [28].

Researchers have characterized Pi transport in osteoclast-like cells derived from RAW264.7 cells after treatment with receptor activator of NF-κB ligand (RANKL). Osteoclast cell differentiation was confirmed by tartrate-resistant acid phosphatase (TRAP) and calcitonin receptor (CTR) staining. The addition of K_2_HPO_4_ induced a slight decrease in intracellular pH. The results suggested that H^+^ flowed into osteoclast-like cells along with Pi. The osteoclast-like cells that were exposed to bone particles showed an increase in H^+^-dependent Pi transport [11]. One possible explanation for these findings is that HCl secretion by osteoclasts produces Ca^2+^, water and phosphate from the CaPO_4_ salt hydroxyapatite. Osteoclast-like cells with a Pi transport system capable of H^+^-dependent stimulation at acidic pH are necessary for bone resorption or production of the massive amounts of energy required by V-type ATPase for acidification of the extracellular environment [11].

### 2.3. The Pi Transport System in Proximal Renal Tubule

In the kidney, a considerable number of functional nephrons play a significant role in Pi homeostasis, where 75% of the filtered Pi is reabsorbed in the proximal tubule; in the distal tubule, only 10% is reabsorbed, and the remaining 15% is lost in the urine [29]. The proximal tubule is an intensively active region because two-thirds of the glomerular filtrate is reabsorbed as ions, water and other molecules within the lumen of the proximal tubule by brush-border membrane microvilli (BBM). The molecules absorbed by the BBM pass into the blood plasma through the cells of the basolateral membrane (BLM) of the proximal tubule [30].

The electrochemical potential of sodium across the BBM provides the driving force for intracellular phosphate accumulation [31]. In renal membranes from the rat kidney cortex, a sodium-phosphate cotransport system localized in the BBM and sodium-independent Pi transport through the BLM have been demonstrated [32].

Phosphate is taken up by the BBM by a sodium–phosphate cotransporter (usually as 2Na^+^ −HPO_4_^−2^), which has a high affinity for Pi (0.1–0.2 mM) and concentrates phosphate in the cytosol [32]. It is speculated that phosphate exits across the BLM by moving down an electrical gradient; as the cytosolic free phosphate concentration is approximately 1.0 mM and the plasma Pi concentration is 2.5–3.0 mM, the transmembrane voltage plays an essential role in BLM transport [10]. Azzarolo et al. analyzed BLM vesicles isolated from porcine kidneys and demonstrated that Pi transfer across the cytoplasm at the basolateral cell side is facilitated by a form of sodium-independent phosphate transport that is specific for phosphate with low affinity (Figure 1 and Table 1) [10].

### 2.4. Pi Uptake by Capillaries of the Blood–Brain Barrier

The BBB restricts the free diffusion of nutrients, hormones and pharmaceuticals between the blood and brain in either the blood-to-brain direction or the brain-to-blood direction [33]. The capillaries of the brain are formed by a specialized endothelium, the function of which is to regulate the movement of solutes between the blood and brain. The concentration of inorganic phosphate in the interstitial fluid of the brain is maintained between 0.5 and 1.0 mM [14].

In bovine brain capillaries, Pi transport is a high-affinity process not regulated by sodium and is not coupled to cations for the translocation of phosphate across capillary membranes. The uptake of phosphate in the capillaries of the BBB reflects transport via a high-affinity system (Table 1) [14].

In addition, Pi uptake has been found to be sensitive to inhibition by arsenate and phosphonoformic acid (PFA). Phosphate transport by isolated capillaries was found to be partially inhibited by inhibitors of anion exchangers, DIDS (4,4′-diisothiocyanatostilbene-2,2′-disulfonate), SITS (4-acetamido-4-isothiocyanostilbene-2,2-disulfonate), and competitively inhibited by low concentrations of various anions, pyruvate, acetate, citrate, glutamate and sulfate, which are subsequently metabolized by the cell in the Krebs cycle to produce ATP [14]. No correlation was shown suggesting the contribution of NaPi-I to Pi transport/anion exchange. Together with these results, Na^+^-independent Pi transport is concluded to be an anion exchanger (Figure 1). Such transport would provide phosphate ions to the cell to supply energy demands and phosphorylation processes and help maintain a low concentration of inorganic phosphate in the brain interstitial fluid [14].

For the same group, it was examined whether BBB phosphate transport is influenced by hormonal and nonhormonal stimuli. Three distinct pathways involving various receptors and second messengers were identified, wherein (1) the stimulation of adenylate cyclase decreases Pi transport, (2) the activation of phospholipase C stimulates Pi transport and (3) the stimulation of tyrosine kinases reduces Pi uptake [34].

## 3. High Phosphate and Pi Transport Mechanisms and Cancer Promotion

### 3.1. Extracellular Pi and Tumorigenesis

In 2007, Pi was shown to be a limiting factor for biological growth; it is a limiting factor mainly because it is one of the fundamental elements necessary for the synthesis of nucleic acids, such as DNA and RNA [35]. Elser et al. published a mathematical calculation based on the growth rate hypothesis (GRH), which predicts a three-quarter reduction in tumor size if the patient halves phosphate intake [35].

Because Pi is an essential nutrient for energy metabolism and because high levels of Pi promote signaling aberrations in tumor cells, much research has focused on the association between hyperphosphatemia and cancer development [17,36]. A clinical study by Papaloucas et al. showed that patients diagnosed with head, neck, lung and cervical cancer had a high serum Pi concentration of 2.52 (±0.72), a concentration twice as high as that in healthy patients without cancer: 1.09 (±0.19) mM [37]. However, the authors of that study did not clarify whether the increase in serum phosphate was a cause or consequence of the disease [37].

Similar to cytokines and various growth factors, phosphate can induce the growth of cancer cells through various growth-promoting signals [37]. In skin cancer cells (JB6), high concentrations of Pi (3 mM Pi) have been found to promote cell transformation and stimulate cell proliferation by activating the N-Ras signaling pathway ERK1/2 [16]. However, by adding an inhibitor of Pi transport (PFA or foscarnet), activation of the N-Ras pathway was blocked at high Pi concentrations. In a mouse model of tumorigenesis induced by a highly carcinogenic chemical agent (7,12-dimethylbenz (a) anthracene12-O-tetradecanoylforbol-13-acetate), administration of a high-Pi diet (3.25 ± 0.58 mM Pi) versus a normal-Pi diet (2.17 ± 0.19 mM Pi) accelerated the induction of papilloma formation [16].

In a mouse model of lung cancer induced by K-rasLA1 mutation, researchers observed that the incidence of lung tumors and tumor diameter were larger in mice fed a high (1.0%)-Pi diet for four months than in those fed a normal (0.5%)-Pi diet for four months [38]. In addition, the same group found that an excessive-Pi diet or Pi restriction can decrease the expression of phosphatase and tensin homologue (PTEN), activate the Akt pathway and increase the expression of the NaPi-IIb transporter at high Pi [39].

### 3.2. Na^+^-Independent Pi Transport Mechanisms in Breast Cancer

In some studies of breast cancer, a role of the sodium-dependent Pi transporter (NaPi-IIb) in tumorigenesis has been proposed [40,41,42]. Russo-Abrahão et al. reported higher Na^+^-dependent Pi transport in MDA-MB-231 cells than in other types of breast cancer cells (MCF-7 and T47-D) [42]. In addition, Na^+^-dependent Pi transport presented Michaelis–Menten kinetics (Km = 0.08 mM; high affinity) [42].

Because the serum Pi range reaches up to 1.2 mM [37], one year later, it was demonstrated that there is another manifestation of Michaelis–Menten kinetics of Km = 1.38 ± 0.16 mM, indicating low-affinity Pi transport acting in a sodium-independent manner [15]. In addition to the lack of dependence of sodium, H^+^-dependent Pi transport stimulated by acidic conditions was verified, suggesting that protons present in the extracellular environment might be transported together with inorganic phosphate by an H^+^-dependent Pi cotransporter [15]. Consistent with these observations, FCCP (an H^+^ ionophore), bafilomycin A1 (an inhibitor of vacuolar H^+^-ATPase) and SCH28080 (an H^+^, K^+^-ATPase inhibitor), which deregulate the intracellular levels of protons, inhibited H^+^-dependent Pi transport. Notably, no effect was observed when anions and anion-carrier inhibitors (DIDS) were tested [15].

Recently, in mammary gland tumors of mice, an acidic region and a high concentration of Pi in the tumor microenvironment (1.8 ± 0.2 mM) compared to that in normal mammary gland (0.84 ± 0.07 mM) were identified as markers of tumor progression [43]. Decreased Na^+^-dependent Pi transport and NaPi-IIb expression in the presence of 2 mM Pi were observed, concomitant with an increase in H^+^-dependent Pi transport [15]. These observations led the researchers involved to suggest the occurrence of a compensatory mechanism for Pi transport in situations where the transport of Na^+^-dependent Pi is compromised [15]. The excess Pi might be able to compensate for the energetically expensive biochemical features of tumor cells. The H^+^-dependent Pi transporter could confer on tumors a biological advantage by endowing cells with the ability to incorporate extra Pi even when sodium-dependent Pi transport is saturated by high extracellular Pi in the tumor microenvironment (approx. 2 mM) [15].

### 3.3. Na^+^-Independent Pi Transport and Metastasis

Based on the growth rate hypothesis, it has been hypothesized that metastasis establishes and forms secondary tumors at organs with higher Pi content than that of the organ containing the original tumor [44]. Lin et al. sought to evaluate the effects of elevated Pi on metastasis and angiogenesis in lung cancer cells (A549) and breast cancer cells (MDA-MB-231). They observed that elevated Pi enhanced cell migration and the expression of angiogenic markers such as VEGF, osteopontin and “Forkhead box protein C2”, a transcription factor related to vasculogenesis and angiogenesis [45].

In a study of breast cancer, a sodium-dependent Pi transport inhibitor (PFA) was found to yield an approximately 50% inhibition of cell adhesion and migration, suggesting the participation of this Pi transporter in cellular motility processes [42]. Recently, Lacerda-Abreu et al. demonstrated the inhibition of the Pi H^+^-dependent transporter by phosphonoacetic acid (PPA), which is able to inhibit cell adhesion and migration by approximately 40% [15].

In fact, these two studies show that the two Pi transporters are important for metastatic processes; however, the biochemical behavior of these transporters differs, and when the Pi concentration in the tumor microenvironment changes, the roles of these Pi transporters in tumor processes might change as well (Figure 2). For example, Lacerda-Abreu et al. used cells that migrated through the Transwell membrane for their Pi transport assays. The migrating cells displayed higher H^+^-dependent Pi transport activity than nonmigrating cells. In contrast, there was no difference between these two conditions when Pi transport activity was measured at 100 μM Pi, a suitable condition for high-affinity Na^+^-dependent Pi transporters [15].

The migration cascade and metastatic invasion are regulated by a multistage process called the epithelial–mesenchymal transition (EMT), in which transformed epithelial cells acquire mesenchymal characteristics, including motility and invasiveness [46]. During EMT, epithelial cells lose epithelial markers (E-cadherin, occludin and cytokeratins) and begin to express mesenchymal markers (vimentin and fibronectin) [46]. Triple-negative breast cancer cells, such as MDA-MB-231 cells, have an EMT morphology (presenting low expression of E-cadherin and high expression of vimentin) and a greater invasive capacity than luminal breast cancer cells (MCF7 and T47-D cells) [15,46]. MDA-MB-231 cells grown in the presence of phosphonoacetic acid (PAA) for 24 h present induced expression of E-cadherin. These observations strongly suggest that when H^+^-dependent Pi transport is inhibited, MDA-MB-231 cells could revert from a mesenchymal phenotype to an epithelial phenotype and consequently exhibit low migratory capacity [15].

### 3.4. Pi Transport Stimulated by [H^+^] in Ehrlich Ascites Tumor Cells

In Ehrlich ascites tumor cells, Na^+^-dependent Pi transport plays a principal role in the maintenance of intracellular Pi concentration. However, a Na^+^-independent component of Pi transport comprises approximately 12% of the total Pi flux [47]. Therefore, Na^+^-dependent and Na^+^-independent Pi transport processes appear to involve, at least functionally, different transporters.

Furthermore, H^+^-stimulated Na^+^-independent Pi transport with saturation kinetics with respect to [H^+^] has been observed. Additionally, it appears that the stimulation of Pi Na^+^-independent transport by H^+^ decreases the intracellular pH (below approximately 6.5), which affects the inhibition of Pi Na^+^-dependent transport, consistent with the interaction of H^+^ with an intracellular site that regulates Na^+^-dependent Pi transport [47].

## 4. Disease Development Related to Hyperphosphatemia and Hypercalcemia

In general, Pi homeostasis and the regulation of renal and intestinal Pi transport are crucial for the normal functioning of human organs. For example, when the renal and intestinal transport of Pi is compromised, hypophosphatemia is promoted, which leads to the dysfunction of several organ systems, including the musculoskeletal system. On the other hand, the dysregulation of the renal and intestinal transport of Pi can also result in hyperphosphatemia, leading to impaired cardiovascular function and soft tissue calcification [5].

Hyperphosphatemia is defined as a serum phosphorus concentration >4.5 mg/dL (1.45 mM). It is a major cause of morbidity and mortality in patients with chronic kidney disease (CKD) and can also be a cause of acute kidney injury (AKI). A decline in renal function leads to phosphate retention, high levels of parathyroid hormone (PTH) and low levels of 1.25-dihydroxy vitamin D [48]. The most common clinical manifestation of hyperphosphatemia is hypocalcemia due to the precipitation of calcium phosphate in soft tissue, which can lead to clinical manifestations of hypocalcemia [7].

### 4.1. Vascular Smooth Muscle Calcification

Vascular calcifications (VCs) are actively regulated biological processes that involve cellular ossification and the participation of factors that either promote or inhibit the organized deposition of hydroxyapatite in vascular smoothing [49]. Susceptibility to VC is genetically determined and actively regulated by many factors. One of these factors is hyperphosphatemia, which promotes VC and is a nontraditional risk factor for cardiovascular disease mortality in end-stage renal disease patients [50].

Vascular smooth muscle cells (VSMCs) in rats were kinetically characterized for phosphate transport. VSMCs exhibit both Na^+^-dependent and Na^+^-independent Pi transport components with similar kinetic behavior and high affinity (Table 1). Both components contribute almost equally to the total uptake of Pi by VSMCs. These kinetic characteristics are very relevant to VC as they are associated with increasing levels of serum phosphate due to hyperphosphatemia, consistent with the theory that Pi influx works as an exact sensor of calcifying conditions (Figure 3) [13].

A study of Pi transport in rat VSMCs revealed that the sodium-independent Pi uptake system is competitively inhibited by sulfate, bicarbonate and arsenate and weakly inhibited by DIDS, SITS and phosphonoformate. These findings indicated that the Pi transport system is most likely coupled to the exit of anions [51]. In addition, an exit pathway from the cell that is partially chloride-dependent and unrelated to the known anion exchangers expressed in VSMCs has been shown to be resistant to DIDS/SITS [51].

To clarify which transporter is responsible for sodium-independent Pi transport, several genes were silenced (SLC4a2, SLC4a3, SLC4a7, SLC26a2, SLC26a6, SLC26a8, SLC26a10 and SLC26a11) using siRNA transfections. None of the interfering transporters were found responsible for a significant part of the sodium-independent influx or efflux of Pi in VSMCs [51].

### 4.2. Crystal Formation in Articular Cartilage and Osteoarthritis Development

Many joint diseases, such as osteoarthritis, are characterized by the eventual destruction of the articular cartilage. Some arthritic conditions are associated with the deposition of crystals, such as hydroxyapatite crystals, in the synovial fluid of articular cartilage of the joint. These crystalline materials include calcium pyrophosphate dihydrate and other forms of calcium phosphate [52]. Calcium and inorganic phosphate are taken up by matrix vesicles (MVs) derived from chondrocytes to form hydroxyapatite crystals, which propagate on collagen fibrils to mineralize the extracellular matrix [53].

Although previous studies of phosphate transport in articular chondrocytes have described high-affinity Pi transport mediated by a PiT system [54], inorganic phosphate transport has been identified in bovine articular chondrocytes, and it appears that high-affinity Na^+^-independent processes contribute to Pi uptake [12]. Regarding Pi transporter inhibitors, PAA and arsenate exhibit low affinities for the Na^+^-dependent component, but markedly higher affinities for the Na^+^-independent component (Table 1) [12]. In addition to chondrocytes, which uptake Pi and Ca^2+^, matrix vesicles have Pi and Ca^2+^ transporters [20]. Solomon et al. characterized the Pi transporters in matrix vesicles (Table 1) [20].

Solomon et al. detected not only high-affinity Pi Na^+^-dependent transport, but also high-affinity Na^+^-independent Pi transport [20]. They proposed that Pi transport systems in chondrocytes or matrix vesicles could contribute to the deposition of minerals in cartilage, thus promoting degenerative joint disorders such as osteoarthritis. Hence, the pathways identified here constitute potential targets for pharmacological intervention to prevent crystal formation and osteoarthritis (Figure 4) [12,20].

## 5. Conclusions

Pi is essential for many, if not all, living organisms because of its roles in several biochemical processes, such as kinase/phosphatase signaling; ATP formation; and lipid, carbohydrate and nucleic acid biosynthesis. Although the importance of sodium-dependent Pi transporters (NaPi type II and NaPi type III) in different tissues and their physiological functions have been demonstrated, an increasing number of studies have reported the presence of another kind of sodium-independent Pi transporter that aids in Pi absorption in intestinal cells and proximal tubule cells as well as reabsorption of Pi by osteoclasts in bone and by capillaries of the BBB.

Sodium-independent Pi transporters also contribute to the accumulation of Pi and other minerals in cartilage and vascular smooth muscles, causing osteoarthritis and VC, respectively. Additionally, Pi is absorbed by a sodium-independent Pi transporter (H^+^-dependent) in breast cancer cells, which enhances the migration and adhesion processes that lead to the development of metastases. The studies included in this review have improved our understanding of Pi homeostasis and have clarified the development of Pi-related diseases.

## Figures and Tables

**Figure 1 ijms-21-09298-f001:**
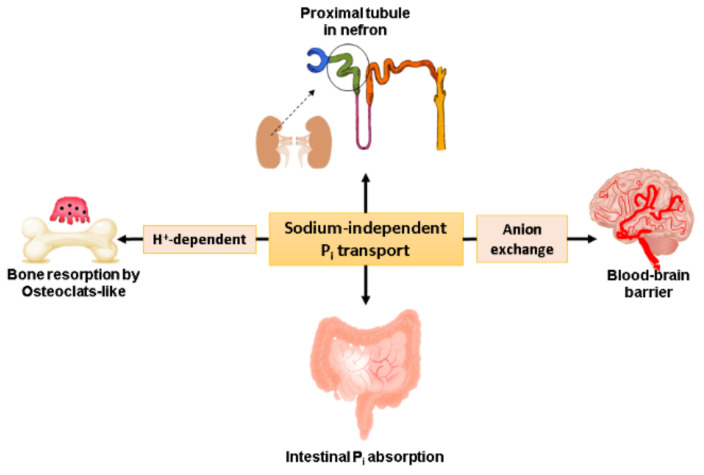
Schematic representation of the roles of sodium-independent Pi transport in Pi homeostasis. Na^+^-independent Pi transport facilitates the absorption of Pi in intestinal cells [9,19] and proximal tubule cells [10] and the reabsorption of Pi in bone by osteoclast-like cells derived from RAW264.7 cells (H+-dependent Pi transport) [11] and in capillaries of the blood–brain barrier (BBB) (anion exchange) [14].

**Figure 2 ijms-21-09298-f002:**
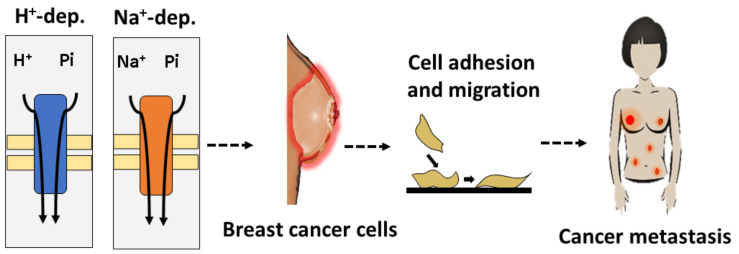
Schematic of the role of sodium-independent Pi transport in the development of breast cancer. Na^+^-independent Pi transport and Na^+^-dependent Pi transport in breast cancer cells promote cell adhesion and migration, which are important for maintaining cancer metastasis [15,42].

**Figure 3 ijms-21-09298-f003:**
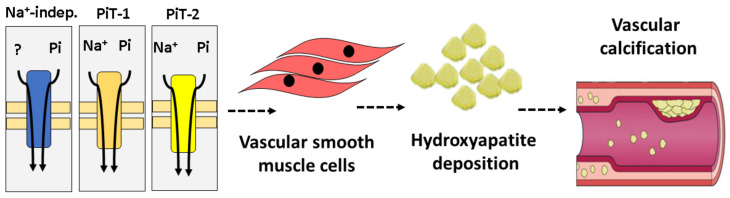
Schematic of the role of sodium-independent Pi transport in the development of vascular calcification (VC). Na^+^-independent and Na^+^-dependent Pi transport (PiT-1 and PiT-2) in vascular smooth muscle cells (VSMCs) contributes to hydroxyapatite deposition in these cells [13,51].

**Figure 4 ijms-21-09298-f004:**
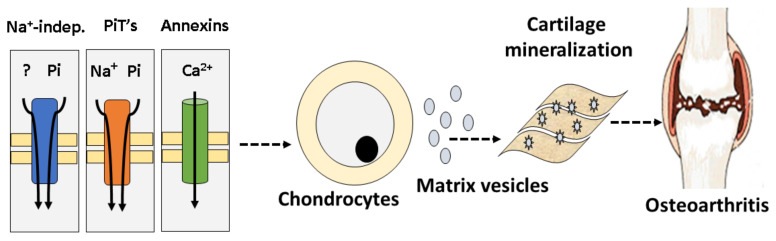
Schematic of the role of sodium-independent Pi transport in the development of osteoarthritis. Na^+^-independent and Na^+^-dependent Pi transport (PiT-1 and PiT-2) and annexins (Ca^2+^ channels) in chondrocytes or matrix vesicles promote cartilage mineralization and consequently osteoarthritis [12,20].

**Table 1 ijms-21-09298-t001:** Kinetic parameters of sodium-independent Pi transport in mammalian cells.

Cell Type or Tissue	Pi Transport	Affinity	K_m_ (mM Pi)	Vmax	Ref
Intestinal Caco2BBE cells at 1 mM Pi	Na^+^-independent proton-activated	High *	0.071 ± 0.020	0.073 ± 0.017 nmol Pi·mg cell protein^−1^·min^−1^	[9]
Intestinal Caco2BBE cells at 4 mM Pi	Na^+^-independent	High *	0.155 ± 0.025	0.849 ± 0.11 nmol Pi·mg cell protein^−1^·min^−1^	[9]
Vascular smooth muscle cells	Na^+^-independent	High	0.10 ± 0.04	180.7 ± 32.8 pmol/mg protein^−1^·min^−1^	[13]
Chondrocytes from articular cartilage	Na^+^-independent	High	0.22 ± 0.07	0.50 ± 0.005 mmol (L cells)^−1^ (10 min)^−1^	[12]
Matrix vesicles from articular cartilage	Na^+^-independent	High	0.16 ± 0.04	0.67 ± 0.04 nmoles (mg protein)^−1^ (min)^−1^	[20]
Osteoclast-like cells	H^+^-dependent	High	0.35	~15 nmol/mg/10 min	[11]
	H^+^-dependent	Low	7.5	~55 nmol/mg/10 min	[11]
Capillaries of the blood–brain barrier	Anion exchanger	High	0.16	0.37 nmol/mg protein/30 s	[14]
Renal basolateral membranes	Na^+^-independent	Low	10.1 ± 1.2	13.6 ± 2.0 nmol (mg protein)^−1^ min^−1^	[10]
Breast cancer cells MDA-MB-231	H^+^-dependent	Low	1.387 ± 0.1674	198.6 ± 10.23 Pi × h^−1^ × mg protein^−1^	[15]

* The affinity was not defined by the authors; however, comparing with other K_m_ values in this table, we suggest that these transporters have high-affinity parameters.

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
