# Peer review of "The Roles of Sodium-Independent Inorganic Phosphate Transporters in Inorganic Phosphate Homeostasis and in Cancer and Other Diseases"

_ijms, 2020, doi:10.3390/ijms21239298_

Round 1

Reviewer 1 Report

The present review approaches the state of art in the field of sodium-independent inorganic phosphate transporters and their implications in Pi homeostasis as well as some pathologic states like cancers, vascular smooth muscle cell calcification and osteoarthritis.

The authors will find hereinafter my comments and suggestions:

  1. Although I am not a native English speaker, in my opinion a critical issue of the manuscript is the inappropriate use of the English language. E.g, “Actually, this uptake exhibits several unique characteristics of the transport mechanisms: saturation, pH dependence, inhibition by putative substrates and decreased transport by inhibitors and has a significant role in Pi homeostasis along with sodium-dependent Pi transporters” (rows 58-60); “The molecules absorbed by the BBM pass into the blood plasma through the basolateral membrane of the cells of the basolateral membrane (BLM) proximal tubule [29]” (rows 148-150) Do you mean basolateral membrane of the proximal tubule cells?; “Phosphate is taken up by the BBM by a sodium-phosphate cotransporter, usually as 2Na+-HPO4-2 has a high affinity, on the order of 0.1-0.2 mM, and concentrates phosphate in the cytosol [31]” (rows 155-156), etc.
  2. Likewise, there are inaccurate terms, i.e. “Pi bone”, “the transport of Pi grown..”, “strongly CaPO4 salt hydroxyapatite”, “rate of biological growth” etc.
  3. The authors wrote: “Pi transport in osteoclast-like cells derived from RAW264.7 cells was characterized by treatment with a stimulator of osteogenesis, RANKL (Figure 1)” (raw 134-135). Figure 1 does not denote what the author state. Besides, the sentence is incomplete because depending on the receptor to which RANKL interacts, it can be induced either osteogenesis or osteoclastogenesis.
  4. Figure 2 consists of three figures A, B and C. I suggest the authors to show individually these three figures in the corresponding place of the manuscript (where they are described). Otherwise the authors must change in the figure caption the reference number for [50] to [46] and for [54] to [47], and so on, because such a disruption in the reference order is confusing to the reader.
  5. The authors quoted the references for several times within the main text as “author name and colaborators”. E.g. Candeal and collaborators (2014), Shirazi-Beechey and collaborators (1986), Lin, McKinnon and other collaborators (2015). Please quote as “Candel et al.” and without the year of publication because the reference is numerically indicated and the year is shown in the References list.
  6. Minor comments: a) Typographical error - Fig. 1: “Sodium-independente” instead of “Sodium-independent”; b) The authors are advised to write the references according to IJMS reference stilling. For instance, there are references where not all authors are mentioned (refs. 5, 36-38, etc), with abbreviated journal name (refs. 20, 39). Ref. 54 - Please delete ”b” in case of “2007b”

Author Response

Reviewer 1

- Major comments:

Question 1: Although I am not a native English speaker, in my opinion a critical issue of the manuscript is the inappropriate use of the English language. E.g, “Actually, this uptake exhibits several unique characteristics of the transport mechanisms: saturation, pH dependence, inhibition by putative substrates and decreased transport by inhibitors and has a significant role in Pi homeostasis along with sodium-dependent Pi transporters” (rows 58-60); “The molecules absorbed by the BBM pass into the blood plasma through the basolateral membrane of the cells of the basolateral membrane (BLM) proximal tubule [29]” (rows 148-150) Do you mean basolateral membrane of the proximal tubule cells?; “Phosphate is taken up by the BBM by a sodium-phosphate cotransporter, usually as 2Na+-HPO4-2 has a high affinity, on the order of 0.1-0.2 mM, and concentrates phosphate in the cytosol [31]” (rows 155-156), etc.

Response: We are sorry for the unpleasant situation, in fact, we are surprised about this problem, since the text passed the AJE (American Journal Experts) correction before this submission (certificate attached at the end of this document). As suggested, we modified the text and improved English writing (rows: 58, 59, 155,156).

About the sentence “The molecules absorbed by the BBM pass into the blood plasma through the basolateral membrane of the cells of the basolateral membrane (BLM) proximal tubule [29]”: Yes, we mean basolateral membrane of the proximal tubule cells and this sentence was corrected in the new manuscript (rows 148-150)

Question 2: Likewise, there are inaccurate terms, i.e. “Pi bone”, “the transport of Pi grown..”, “strongly CaPO4 salt hydroxyapatite”, “rate of biological growth” etc.

Response: We corrected the inaccurate terms in the text (rows 16, 79, 139, 190)

Question 3: The authors wrote: “Pi transport in osteoclast-like cells derived from RAW264.7 cells was characterized by treatment with a stimulator of osteogenesis, RANKL (Figure 1)” (raw 134-135). Figure 1 does not denote what the author state. Besides, the sentence is incomplete because depending on the receptor to which RANKL interacts, it can be induced either osteogenesis or osteoclastogenesis.

Response: We agree with the reviewer, and corrected the sentences (rows: 76,77, 133-135).

Question 4: Figure 2 consists of three figures A, B and C. I suggest the authors to show individually these three figures in the corresponding place of the manuscript (where they are described). Otherwise the authors must change in the figure caption the reference number for [50] to [46] and for [54] to [47], and so on, because such a disruption in the reference order is confusing to the reader.

Response: We agree with the reviewer and separated the figure 2 into new figures 2, 3 and 4 (rows: 73, 257, 262-265, 314-318, 350-354).

Question 5: The authors quoted the references for several times within the main text as “author name and colaborators”. E.g. Candeal and collaborators (2014), Shirazi-Beechey and collaborators (1986), Lin, McKinnon and other collaborators (2015). Please quote as “Candel et al.” and without the year of publication because the reference is numerically indicated and the year is shown in the References list.

Response: As suggested, we modified the references throughout the text (rows: 88, 116, 191, 196, 217, 244, 251, 257).

-Minor comments

Question 6:

  1. a) Typographical error - Fig. 1: “Sodium-independente” instead of “Sodium-independent”;

Response: As suggested, we modified the word in the new manuscript (new figure 1).

  1. b) The authors are advised to write the references according to IJMS reference stilling. For instance, there are references where not all authors are mentioned (refs. 5, 36-38, etc), with abbreviated journal name (refs. 20, 39).

Response: As suggested, we wrote the references according to IJMS reference stilling (rows: 387, 436, 475, 476, 479, 480, 483, 488, 492).

  1. c) Ref. 54 - Please delete ”b” in case of “2007b

Response: As suggested, we deleted ‘b’ in the reference list (rows: 407, 527).

The original manuscript (OM) “The sodium-independent inorganic phosphate transporters: inorganic phosphate homeostasis, cancers and other diseases” has been revised according to the Reviewers’ criticisms and suggestions. The net result is a revised and new manuscript (RM).

We really appreciate the critical appraisal from the Reviewers and hope that, in the present form, the manuscript is acceptable for publication in International Journal of Molecular Science.

            Thank you for your consideration.

Reviewer 2 Report

The presented manuscript is in the scope of the journal and it is relatively easy to read and and understand to the wide readership. I recommend to the editorial board to accept the manuscript with minor revision, because I did not found graphic illustration of figure 2 in the text of manuscript. In this regard, I propose to the authors to make a revision of their manuscript and to include the graphic.

Author Response

Reviewer 2:

Question: The presented manuscript is in the scope of the journal and it is relatively easy to read and and understand to the wide readership. I recommend to the editorial board to accept the manuscript with minor revision, because I did not found graphic illustration of figure 2 in the text of manuscript. In this regard, I propose to the authors to make a revision of their manuscript and to include the graphic.

Response: Regarding the figure 2, as suggested by reviewer 1, we separated in figure 2, 3 and 4, included in the revised manuscript (rows: 73, 257, 262-265, 314-318, 350-354).

The original manuscript (OM) “The sodium-independent inorganic phosphate transporters: inorganic phosphate homeostasis, cancers and other diseases” has been revised according to the Reviewers’ criticisms and suggestions. The net result is a revised and new manuscript (RM).

We really appreciate the critical appraisal from the Reviewers and hope that, in the present form, the manuscript is acceptable for publication in International Journal of Molecular Science.

            Thank you for your consideration.

Round 2

Reviewer 1 Report

The authors appropriately responded to my comments

but I am still concerned about the correct use of the English language.  

Author Response

Reviewer 1

- Minor comments: The authors appropriately responded to my comments but I am still concerned about the correct use of the English language.

Response: As required, the manuscript was revised once again and the use of the English language, grammar, punctuation, spelling, and overall style have been corrected by one or more of the highly qualified native English-speaking editors at AJE. The certificate for this new revision is attached to this document. Besides, the title of this work has also been changed due the English language correction.
